# Bioinspired Pd-Cu Alloy Nanoparticles as Accept Agent for Dye Degradation Performances

**DOI:** 10.3390/ijms232214072

**Published:** 2022-11-15

**Authors:** Shiyue Chen, Yujun Yang, Mingjun Zhang, Xiaohong Ma, Xiaoxiao He, Teng Wang, Xi Hu, Xiang Mao

**Affiliations:** 1State Key Laboratory of Ultrasound in Medicine and Engineering, College of Biomedical Engineering, Chongqing Medical University, Chongqing 400016, China; 2Chongqing Key Laboratory of Biomedical Engineering, College of Biomedical Engineering, Chongqing Medical University, Chongqing 400016, China; 3Chongqing Medical Laboratory Microfluidics and SPRi Engineering Research Center, Key Laboratory of Laboratory Medical Diagnostics, Ministry of Education, Department of Laboratory Medicine, Chongqing Medical University, Chongqing 400016, China; 4State Key Laboratory of Multi-Phase Complex Systems, Institute of Process Engineering, Chinese Academy of Sciences, Beijing 100190, China

**Keywords:** Pd-Cu NPs, hydrophilic, biocompatibility, catalytic agent, dye degradation

## Abstract

Dye degradation is a key reaction in organic decomposition production through electron donor transferring. Palladium (Pd) is the best-known element for synthesis Pd-based catalyst, the surface status determines the scope of relative applications. Here we first prepare Pd-Cu alloy nanoparticles (NPs) by co-reduction of Cu(acac)_2_ (acac = acetylacetonate) and Pd(C_5_HF_6_O_2_)_2_ in the presence of sodium borohydride (NaBH_4_) and glutathione (GSH). The obtained Pd-Cu is about ~10 nm with super-hydrophilicity in aqueous mediums. The structural analysis clearly demonstrated the uniform distribution of Pd and Cu element. The colloidal solution keeps stability even during 30 days. Bimetallic Pd-Cu NPs shows biocompatibility in form of cell lines (IMEF, HACAT, and 239 T) exposed to colloidal solution (50 µg mL^−1^) for 2 days. It shows the catalytic multi-performance for dye degradation such as methyl orange (MO), rhodamine B (RhB), and methylene blue (MB), respectively. The as-synthesized nanoparticles showed one of the best multiple catalytic activities in the industrially important (electro)-catalytic reduction of 4-nitrophenol (4-NP) to corresponding amines with noticeable reduced reaction time and increased rate constant without the use of any large area support. In addition, it exhibits peroxidase-like activity in the 3, 3′, 5, 5′-Tetramethylbenzidine (TMB) color test and exhibit obvious difference with previous individual metal materials. By treated with high intensity focused ultrasound filed (HIFU), Pd-Cu NPs might be recrystallized and decreased the diameters than before. The enhancement in catalytic performance is observed obviously. This work expedites rational design and synthesis of the high-hierarchy alloy catalyst for biological and environment-friendly agents.

## 1. Introduction

Recently, alloy nanoparticles (NPs) have attracted widely attention because of their particular catalytic, electronic, optical and other integrated properties [1,2]. Also, under various utilization in different fields, such as Pt-based catalyst in fuel battery [3,4,5], oxidation reduction reaction (ORR), hydrogen evolution reaction (HER) [6,7,8,9], and biocompatibility integration [10,11]. But, due to the scarcity and high cost of platinum itself, partially replacing Pt by earth-abundant transition metals had recently emerged as one of the most effective strategies to optimize the use of Pt, such as Pd-based alloys [12,13,14,15]. It exhibits great comprehensive performance mainly used in oxygen reduction reaction (ORR), ion surfacing, spray welding and others. Another biometallic alloy catalyst, such as Au-based alloy NPs, Au-based alloy NPs play a significant role in science and technology, but Au-based alloy NPs have been limited to the oxidation of CO, benzyl alchol and propene, and Au is expensive, difficult to preserve and complicated to prepare, so it is not a good choice for alloy preparation [16,17,18,19,20,21,22]. But the synthetic approach is not easy to be realized in regular controlling works. The biggest distinguish of alloys could be determined by components which offers the possible utilization while in characterizations. Ascribed to the requirement of additional functionalization, many alloys (such as FeAu, FePt, PtAu, PdPt bimetallic NPs et al.) are modified or reconstructed in forming hydrophilic building blocks. It is the necessary work for further utilization when alloys were using in electrocatalytic and degradation performance during in environmental and new energy conversion [23,24,25,26]. The basic principle in fabricating catalytic agents for hydrophilic dyes degradation in water mediums, hydrophilic status illustrates remarkable possibility and being considered as one important factor to be alloyed and surface structures [27,28]. It might branch out applied fields such as the investigation of internal electronic transfer and synergetic functions while using hydrophilic alloys in works. Surely, the hydrophobic interface exhibits completely opposite state, it prevents alloys contact target molecules between catalytic agents and target molecules [29]. Without any enhancement about catalytic performance, there implies the necessary about hydrophilic status, it meant high stability (solution status) and mono-disperse in colloidal solutions [30].

Ordinarily, the catalytic performance is mainly depended on size distribution, surface modification, compositions and crystalline parameters [31,32,33]. Among Pd based bimetallic alloys, it can keep the functional properties of each component and provide synergistic effects, which resulting in various characteristics [34,35,36]. The alloying procedures could provide a unique strategy for customizing the geometric and electronic structure in forming catalyst surface [12,34,37]. Pd-based alloys are easy to produce and widely available, which is an acceptable choice for alloy preparation. And the catalyst phenomenon also shows oxidation-reduction reactions (ORR) activities and present a cost-effective alternative to previous metal candidates [38,39]. As mentioned in previous works, a series of Pd based alloys were fabricated in solvothermal method, it exhibits narrow size distribution and mono-dispersion in organic solvents [40,41,42]. In order to be well documented as efficient catalyst or biocompatible metal enzymes [43,44], the bimetallic NPs can be modified as integrating with commercial carbon powder and ligands exchanges in surface rebuilt procedures [45,46,47]. As one particular instance, the hydrophilic Pd-Cu alloy NPs can be fabricated but its status of dispersion and stability were poorly due to the complex procedures of hydrophobic to hydrophilic status. Among them, it used hydrophilic molecules or polymers for “secondary” modification in forming catalyst. During this, it could obtain new surface modification, resulting in more stable alloy NPs, but it may reduce the original catalytic activity partly [42,48,49,50,51]. Additionally, it seems that the consistent size value of alloys should be one significant factor in determining catalytic performance. Certainly, there have requirements toward maintaining the diameter of NPs steadily without using any further modification works, it can greatly increase the exposed sites and increase catalytic performance [52,53].

Pd-Cu alloy NPs are one of representative particles in Pd bimetallic alloys with multivariate properties. Which can be ascribed to copper ions (cation) could switch between different valence states (+1, +2), in which, Cu^+^ could lose an electron to become Cu^2+^, it could gain an ion to become Cu^+^ in controlling electroreduction and peroxide-like activity works [54,55]. Conventionally, the synthesis of hydrophilic NPs requires two steps: firstly, hydrophobic NPs were obtained by reducing metal precursors in organic solvents under high temperature; secondly, the modification of NPs’ surface by using hydrophilic molecules (polymers, amino acid, and others). Till now, there have not works refer to Pd-Cu alloy NPs were measured in metallic enzyme-like activities yet. The hydrophobic Pd-Cu NPs are still under chaotic describes during in dye degradation works, it directly proved that hydrophobic state of Pd-Cu alloy NPs (low polarity), which is difficult to show its best alloy properties under hydrophilic environment [56,57,58]. As organic enzyme, the expanded alloys’ characteristics should be endowed with highly activities in further developments. Such as it mimics the enzyme-like activity of horseradish peroxidase (HRP), the great interest ascribed to their application in bioanalytical and clinical chemistry [59,60,61,62,63,64]. Thus, it realizes the promotion in their catalytic properties through constructing nanozymes with different compositions and morphologies [59,60,65,66,67]. In this case, Pd-Cu alloy NPs can work as one metallic nanozymes, which possess super hydrophilic status in water mediums with consistent stability, but also it has much higher catalytic activity and wide selectivity rather than HRP only. By concluding above reasonable requirement, the designed Pd-Cu alloy NPs should be synthesized with super-hydrophilic characteristic, but also maintain multifunction in catalytic reaction.

It is well known that there are many synthetic methods to prepare alloy NPs. For example, Chng et al. developed a fast and facile approach for preparation of hjghly monodisperse AuAg alloy NPs using supramolecular solids M (I)-alkanethiolates (M = Au and Ag) as metal sources [67]; Kim et al. used an electrodeposition method to prepare Co-Cu alloy catalysts, the compositions of which were controlled by varying the concentration of the Co and Cu precursors in the electrolyte [68]; In Shao et al.’s previous job, the porous/hollow Pd-Cu NPs were fabricated by selectively dissolving a less noble metal, Cu, using an electrochemical dealloying process [69]. Wang et al. prepared Fe_3_O_4_@PEI@Ag alloy NPs via a chemical co-precipitation method [70]. Due to the complex preparation method before, in this paper, we used a convenient one-step process to synthesize alloy NPs at RT.

Herein, we reported a one-step synthetic approach for making super-hydrophilic Pd-Cu alloys by using glutathione (GSH) as the surfactant in water medium at RT in Figure 1. The achieved Pd-Cu alloy NPs could show high stability and mono-dispersion even the colloidal solution was kept 30 days at ambient conditions. It clearly proved the super-hydrophilic surface status, which ensures water-soluble and stable properties. By controlling the proportion of precursors, it was easily to synthesize effective catalysts with different sizes and shapes for biocompatible and catalytic actives. The biocompatibility was realized by using three kinds cells (IMEF, HACAT, 239 T cells) lines and the cell viability can reach to 98.8%. Furthermore, the synthesized Pd-Cu alloy NPs showed the distinguished catalytic activity for the reduction of nitro-aromatic compounds and degradation of organic dye molecules. The peroxidase-like activity of Pd-Cu alloy NPs in the 3,3′,5,5′-Tetramethylbenzidine (TMB) color test and showed obvious difference with previous individual metal particles. The structural analysis clearly demonstrated the uniform distribution of Pd and Cu consists, which indicating a homogeneous characteristic in overall the entire system. By treated in high intensity focused ultrasound filed, the enhancement in catalytic performance can be ascribed to the size decreasing and surface area increasing.

## 2. Results and Discussion

**Structure, morphology, optical property and biocompatibility characterizations.** The preparation procedure and proposed catalytic applications of Pd-Cu alloy NPs are shown in Figure 1. It conveys the synthetic approach could be facilely applied for forming hydrophilic status. There have same metal-organic intermediate products while glutathione (GSH) linking with Cu^2+^ and Pd^2+^ in water-based mediums. Initially, it is observed as precipitations among the cross-link phenomenon, which is similar as our previous works [60]. By adding the sodium borohydride as the reducer, the resulted Pd-Cu appeared along with colloidal solution gradually become more darkness, which implied alloying fabrication was achieved during in growth processes. Refers to products yield, it alternatively choses Pd_5_Cu_1_, Pd_7_Cu_1_ alloys NPs as primary maintained samples in further works. Also, the size, morphology, and crystal structure are further examined by transmission electron microscopy (TEM images, Figure 1a,d), which represents crystalline characteristics and solubility by using different metal precursors mole ratio (Pd_5_Cu_1_, Pd_7_Cu_1_) and illustrates low magnified TEM images of Pd_5_Cu_1_, Pd_7_Cu_1_ compositions, respectively. The average size distribution can be maintained about ~10 nm, and each of these two different alloys also keeps highly stability in water mediums. Additionally, the high-angle annular dark-field scanning TEM (HAADF-STEM) images and element diffraction analysis (EDS) mapping clearly demonstrate the composition distribution of Pd_5_Cu_1_, Pd_7_Cu_1_ alloys NPs (Figure 1a,d). These resulted images indicated a distribution of Pd-Cu alloy NPs in entire system. By adjusting the synthetic parameters, a series of NPs were prepared such as Pd_1_Cu_2_, Pd_2_Cu_1_, Pd_1_Cu_3_, Pd_3_Cu_1_, Pd_1_Cu_5_, Pd_1_Cu_7_ (as shown in Appendix A). It keeps similar size range as before, but also the final diameter reaches to 5 nm. It conveys one of regular clues about synthesis pure-phased inorganic functional materials in water medium at RT. In the presence of strong reducing agent (BH_4_^−^), palladium and copper atoms can not only exist directly in aqueous solution, but also cause alloying for the continuation of growth process [66]. In order to measure the stability of colloidal solutions, it was characterized by checking the absorption spectra along the different time increasing (Figure 1b,e). It shows that the intensity of absorption would not have changes, but also the physical status (color, mono-dispersion, zeta potential values) keeps same as before, which indicated that Pd-Cu alloys NPs insure certain stability. There have not any vigorous changes appeared before and after 30 days cultivation. The similar phenomenon exhibit in others alloys NPs as shown in Appendix A. On the other hand, it reflects the water-soluble characteristics of Pd-Cu is one essential factor as super-hydrophilic agents.

Powder X-ray diffraction (PXRD) illustrates the crystalline patterns showed similar diffraction peaks, but also could be assigned to face-centered cubic structures (Figure 1c). The peaks of Pd-Cu alloy NPs at 38.2°, 44.2°, and 64.4° were (111), (200), (220), respectively. The peak around 30° was the excess GSH in the alloy NPs. The appeared same diffraction peaks, it indicates both of two samples have the similar crystal structures in growth processes. Alternatively, it implies the relative catalytic performance enhancement along with the Pd increased composition. Fourier transform infrared (FT-IR) is mainly used to achieve non-destructive chemical composition identification of materials at the nanoscale as shown in Figure 1f. In this characterization, the similar bands are observed from 4000 cm^−1^ to 400 cm^−1^, indicating that the main components of each sample have the same chemical structures. The general characteristics of the spectra are common to those of Pd-Cu alloy NPs reported elsewhere [67]. A broad band appears near 3400 cm^−1^, and changes in functional groups containing H- and O- and possible C-C and O-H lead to several possible assignments of the observed bands. The broad band observed near 3400 cm^−1^ may be due to the -O-H stretching of the hydroxyl group expected to be present in the aqueous solution. Because the samples were lyophilized and stored in a desiccator prior to analysis, the dried solid was hygroscopic. 

**The Biocompatibility Measurement.** Due to the designed applications, the Pd-Cu alloy NPs were used to check the biocompatibility for further investigation, which should be documented and comprehensively studied as conventional measurement. The toxicity of nanoparticles was assessed by the viability of cells grown in fusion with different cell lines (239 T, HACAT, IMEF cells) at different concentrations of colloidal solutions, as shown in Figure 2. The cell viability of the four Pd-Cu alloys (mole ratio, Pd:Cu = 1:5, 1:7, 5:1 and 7:1) showed good adaptability to various cells after 2 days of intracellular culture. As a matter of fact, the cell exhibited >98.8% viability at concentrations of up to 50 µg mL^−1^, indicating high biocompatibility. Most of the macrophages incubated with NPs were stained green, implying that the alloy NPs were readily phagocytized by these three cell lines and remained dominant in the cytoplasma for almost 5 days. Similarly, it can be considered non-toxic bimetallic candidate owing to the nature of the surface molecules and its chemical modification (Figure 2b). The fluorescence microscopy images conveyed relative cell integration with NPs, the microscopic observation of the cells further revealed morphological changes. It found that NPs cultivated with varying concentrations had only differed slightly while compared with control group. The observation of NPs treated cells showed that the colloidal stability changed slightly even after the 3 days treatment (Appendix A). Therefore, the Pd-Cu alloy NPs constitute non-toxic candidates for the three types of cell lines.

**The Catalyst Reduction Measurement.** Organic dyes such as methyl orange (MO), Rhodamine B (RhB), methyl green (MG), methylene blue (MB), methyl red (MR) are commonly used in industrial staining and biological labeling applications, which can cause adverse effects on the environment, So, it is recommended that they must be degraded before release into the aquatic environment. In this work, MO, RhB and MB dyes were selected to study the model degradation activity of synthesized Pd-Cu catalysts. The degradation reaction was studied by adding NaBH_4_ to an aqueous solution in the presence of Pd-Cu catalyst, and the reaction kinetics was monitored by measuring optical absorption relative to time in an UV-*vis* spectrometer. To evaluate the catalytic performance of Pd-Cu, MB was selected as first model. Initially, The MB dye shows an absorption peak at 663 nm and 612 nm, which remains unchanged in the presence of NaBH_4_, even up to 30 min, indicated that the reduction process would not occur without catalyst. Details could be seen in the Appendix A. As a representative case of the catalyst (Pd_7_Cu_1_ alloys), the reduction reaction can complete finish within 5 s after the added 0.04 mg catalyst (Appendix A). The first-order kinetics was applied to calculate the catalytic performance of all catalyst. The pseudo-first-order kinetics is suitable one for calculating the reaction rate constant as following:(1)$$\ln\left(C_{t}/C_{0}\right)=\;\ln\left(A_{t}/A_{0}\right)=-\mathrm{kt}\;$$

Here, C_t_ and C_0_ are the absorbance of the MB solution at times t = t and t = 0, respectively, which are equivalent to the concentrations at times t = t (C_t_) and t = 0 (C_0_). The rate constant was measured from linear plots of (C_t_/C_0_) vs. reduction time. K_app_ was determined by each factor from the slope of a linear plot and presented in Tables. These results clearly demonstrated the higher activity of alloy NPs over their monometallic counterparts. Pd_7_Cu_1_ alloy NPs showed the highest reaction rate and effective degradation in whole works. Also, the degradation processes of others proportion were reflected in the supporting information (Appendix A). The effects of MB concentration and catalytic loading were studied by choosing the highest active Pd_7_Cu_1_ alloy NPs as the primary catalyst.

**Catalytic Reduction of MB/RhB/MO Molecules.** Figure 3 represents the absorption spectra of the degradation of MB by NaBH_4_ in the presence of Pd-Cu alloy NPs with different precursor proportions catalyst. The concentration of substrate MB was controlled at 0.07 mM, and the content of catalyst was controlled at 0.01 mg. The influence of different catalysts on the reaction rate was compared, they need 270 s, 240 s, 210 s, 180 s, 150 s, 90 s, 60 s, 40 s, 30 s, respectively. The apparent rate constant was calculated from a linear curve of ln (C_t_/C_0_) over time, as described previously, and its values are shown in Appendix A. Among all alloy components, the Pd_7_Cu_1_ alloy sample showed the highest reaction rate (Kapp = 100 × 10^−3^ s^−1^), which was nearly eight times higher than Pd_1_Cu_1_ (12.9 × 10^−3^ s^−1^) NPs, showing a strong synergistic effect of Pd and Cu alloys. As can be seen from the spectra, MB has a characteristic absorption peak at 612 nm and 663 nm. When the experiments were performed without the Pd-Cu catalyst, it was observed that NaBH_4_ activity was almost inactive for a long period of time [1]. It can be seen that the dye degradation reaction actually provides a surface for the catalyst to adsorb the substrate dye and the reducing molecule of sodium borohydride, and then speeds up the reaction rate by increasing the electron transfer between the reactive molecules. In the present reaction, the concentration of NaBH_4_ was excessive and almost unchanged throughout the reaction system. Therefore, a first-order kinetic approach was used to study the reaction kinetics. The effects of MB concentration and catalytic loading were studied by choosing the highest active Pd_7_Cu_1_ alloy NPs as the primary catalyst (Figure 4). As expected, the dye degradation reaction can be completed in a shorter time by increasing the amount of catalyst. To compare the reaction rates, we calculated the activity parameters (K, s^−1^ g^−1^), as shown in Table 1. From these results, it can be seen that as the concentration of MB increases, the reaction rate decreases, a phenomenon that supports the first-order kinetics of the degradation process. This rate is reduced because the electron transfer process between the NaBH_4_ molecule and the dye molecule on the catalyst surface is slowed down. However, both the reaction rate and the activity parameters increased with increasing catalyst loading, because the increase in catalyst led to an increase in the effective catalytic surface area for the degradation reaction. 

From the analysis of optical property measurement, this outstanding catalytic activity was observed in measuring hydrophilic Pd-Cu alloy NPs. Also, the related result and parameter was showed in Table 1. The exhibited performance was over than other reports, this is mainly due to the following reasons: (a) size effect, The effective surface area of the catalyst is higher for nanoparticles due to their small size. It is generally accepted that the apparent kinetic rate constant is proportional to the total surface area (S) of the nanoparticles [71]; (b) the structure effect, During the formation of alloy NPs, structural defects are generated, which is beneficial to enhance the adsorption of the matrix and reducing agent, thus improving the catalytic properties of alloy NP [72]; (c) electronic effect, the electronegativity difference between Pd (2.20) and Cu (1.90) results in electron rich and electron poor regions on the bimetallic surface. The presence of these regions on the surface of metal alloys facilitates electronic communication between adsorbed molecules [73,74]. The adsorption capacity of alloy NPs was enhanced. Therefore, the alloying process of Pd and Cu will lead to higher absorption and more electron transfer between the substrate molecules. The results showed that the catalyst was catalytically active. In this case, bimetallic Pd-Cu alloy NPs have both electronegative differences and synergistic effects during degradation. This meant that the different metals likewise play a major role in determining the final properties of the alloy. Therefore, the alloy NPs have higher catalytic activity than the pure metals (Pd, Cu).

RhB has a UV absorption peak at 554 nm and MO is at 462 nm. Figure 5a,d represent the UV-*vis* degradation spectra of 0.06 mM RhB with 0.2 mg of Pd-Cu alloy NPs, and 0.04 mM MO with 0.2 mg of Pd-Cu alloy NPs, respectively. The absorption peak of RhB at 554 nm disappeared within 10 s in the presence of NaBH_4_ and catalyst, and the absorption peak of MO is within 8 s, again showing the high activity of our catalyst. As shown in Appendix A, Appendix A represents the UV-*vis* absorbance spectra of degradation of 0.06–0.12 mM RhB by catalyst in 0.2 mg. Appendix A represents the UV-*vis* absorbance spectra of degradation of 0.12 mM RhB by catalyst in 0.2–0.8 mg. Appendix A represents the UV-*vis* absorbance spectra of degradation of 0.04–0.1 mM MO by catalyst in 0.2 mg. Appendix A represents the UV-*vis* absorbance spectra of degradation of 0.1 mM MO by catalyst in 0.2–0.8 mg. The pseudo-first-order kinetic linear simulation curves of ln (C_t_/C_0_) versus reaction time for the catalytic degradation of RhB/ MO is in Figure 5b,c,e,f. The apparent rate constant was calculated from a linear plot of ln (C_t_/C_0_) vs time, shown in Table 1. The rate constant for degradation of RhB was found to be 73.2 × 10^−2^ s^−1^, MO was 53.5 × 10^−2^ s^−1^. The high catalytic degradation activity of the catalyst was discussed in the previous section. 

**Catalytic Reduction of 4-Nitrophenol.** 4-NP is mainly used as an intermediate of fine chemicals such as pesticides, pharmaceuticals and dyes. In this chapter, in order to study the catalytic performance of Pd_7_Cu_1_ alloy NPs, we chose 4-NP for related experiments. The reaction process was recorded by UV—*vis* absorption. In the configuration of the 4-NP solution, we can observe that its color is light yellow, and after the addition of NaBH_4_ the solution becomes deep yellow. This is mainly because the increase in NaBH_4_ leads to an increase in the alkalinity of the solution, which in turn leads to an increase in the 4-NP ion. As in the previous experiments, we first changed the amount of 4-NP by controlling the amount of catalyst (100 μL, 0.2 mg/mL) and NaBH4(100 μL, 0.2 M) constant, as shown in Appendix A, with increasing amount of 4-NP, 0.1 mM degradation could be completed within 4.5 min. Under the condition of controlling the amount of 4-NP (2.5 mL, 0.1 mM) and NaBH_4_ (100 μL, 0.2 M) unchanged, the catalytic reduction reaction could be completed within 2 min with the increase of catalyst dosage. The apparent rate constant of 4-NP reduction was calculated by plotting ln (Ct/C0) vs time (Figure 6), and the apparent rate constant (K_app_) was determined from the slope of the linear plot, as shown in Appendix A.

**The peroxidase-like catalytic activity of Pd-Cu alloy NPs.** In determine the catalytic mechanism of the Pd-Cu alloy NPs, the steady-state kinetic parameters of TMB oxidation in the presence of H_2_O_2_ were determined and the results were placed in a Michaelis-Menten kinetic model. Firstly, we measured the UV-vis absorbance spectra of peroxidase-like activity in different concentrations of the Pd-Cu alloy NPs. As showed in Appendix A, the Pd1Cu7 alloy NPs has the highest peroxidase-like activity; Secondly, by changing the concentration of one substrate and keep other substrates constant, the H_2_O_2_ was 100 mM, adjusted the TMB (1 mM to 10 mM); Thirdly, keep the concentration of TMB at 20 mM and the concentration of H_2_O_2_ at 2 mM to 12 mM (Appendix A). Typical Michaelis-Menten curves of Pd1Cu7 alloy NPs (Figure 7a,b) were obstained for suitable substrate concentrations (TMB and H_2_O_2_). The data were well fitted to the Michaelis-Menten model, resulting in the enzyme kinetic parameters summarized in Appendix A (Michaelis constant, maximum reaction rate, catalytic efficiency and catalytic constant). The Vmax value is a direct measurement of the catalytic of the enzyme. Km is considered to be an indicator of the affinity of the enzyme for the substrate. 

A lower Km value indicates a higher affinity of the alloy nanoenzyme. The result demonstrated that Pd-Cu alloy NPs have peroxidase-like activities and can serve as attractive candidates for peroxidase in form of metalloenzyme types. It conveyed that the oxidation of TMB molecule could occurred Pd-Cu alloy NPs presence of electron donor (H_2_O_2_, TMB to TMBDI). To further analyze the catalytic mechanism, double reciprocal plots of the catalytic mechanism of the initial velocity were measured in Figure 7c,d. It shows double reciprocal plots of initial velocity against one substrate concentration, obtained in a range of concentrations of the second substrate. This finding clearly demonstrates that the catalytic mechanism of the Pd-Cu alloy NPs, in which, catalyst binds and reacts with the first substrate, then releases the grade I products before reacting with the second substrate. 

**The Enhancement in Catalytic Performance of Pd-Cu According to High Intensity Focused Ultrasound Processing.** For the additional attempts in changing or enhancing the catalytic performance of Pd-Cu alloys NPs. The typical treatment of high intensity focused ultrasound field was utilized, such of particularity provided cavitation bubbles and active free radicals in whole water aqueous [75]. The vigorous energy releasing was observed, the theorical temperature could reach to 5000 K at focused central part [76]. It could have higher efficiency for changing the original morphology of functional crystalline at central part (<3.14 × 10^−10^ m^3^). As showed in Figure 8, it completely illustrated that Pd-Cu alloy NPs exhibit relative enhancement of morphological and catalytic performances. Among this, it observed that two states of Pd_5_Cu_1_, Pd_7_Cu_1_ NPs were treated with high intensity focused ultrasound fields (10 MPa, 3000 W and 4000 W). Initially, compared with original information, it mentioned that the relative diameter of Pd-Cu alloy NPs obviously decreased under 10 MPa, 3000 W, and 10 MPa, 4000 W, respectively. As showed in Appendix A, we can see that the size of Pd_5_Cu_1_ alloy NPs decreased to about 3.3 nm. In the same case, the Pd_7_Cu_1_ alloy NPs decreased to 3.1 nm and 3.7 nm under 10 MPa, 3000 W, and 10 MPa, 4000 W in Appendix A, respectively. The digital images of colloidal solutions were shown in Appendix A, respectively. It implied that there have not any changes in Pd-Cu formations under low pressure (0.1 MPa, 3 MPa) and low powers (100 W). It is found that the color of the solution is darker than not treated by ultrasound, and it can be speculated that the shape or structure of NPs has changed. Pd-Cu alloy NPs are based on the state of metal atoms (Pd^0^, Cu^0^) in crystalline characteristics. It experienced the vibration, growth, and constantly gather energy in form of a sharp collapse (≥20,920 J) [77]. The metal atoms (Pd, Cu) could lose electrons at special physical treatment, and each of Pd, Cu can become ion state easily due to their valence state [78]. On the other hand, it has been reported that the sonochemical reduction in the presence of an organic additive proceeds via the following details equations [79,80,81,82]:(2)$$H_{2}O\to *H\left(H^{+}\right)+*\mathrm{OH}\left({\mathrm{OH}}^{-}\right)$$
(3)$$\frac{*H}{*\mathrm{OH}}+\;\mathrm{Absolute}-\mathrm{ethyl}-\mathrm{alcohol}\;\to \;\mathrm{Secondary}\;\mathrm{Radicals}\;+{\;H}_{2}\;$$
(4)Absolute−ethyl−alcohol → *R + Secondary Radicals 
(5)$$\mathrm{Radicals}\;+{\;\mathrm{Pd}}^{2+}+{\;\mathrm{Cu}}^{2+}\;\to \;\mathrm{Pd}-\mathrm{Cu}\;\mathrm{alloy}\;\mathrm{NP}\;$$
where *R denotes a middle material, Equations (2)–(5) indicate the sonochemical formation of the reducing radicals and reductants: *H and *OH are formed from sonolysis of water, secondary reducing radicals and H_2_ are formed from the abstraction reaction of absolute ethyl alcohol and *H or *OH; *R and other secondary reducing radicals are from a pyrolysis of absolute ethyl alcohol and water. Finally, the reduction of Pd-Cu NPs process with various reducing species and involves the complex reaction steps. Pd^2+^ or Cu^2+^ were reduced to Pd or Cu atoms and effected by the power and pressure (Figure 8a). To further study the performance changes of Pd-Cu alloy NPs before and after ultrasound treatment (10 MPa, 3000 W and 10 MPa, 4000 W), we carried out dye degradation experiments. Appendix A showed the UV-*vis* absorbance spectra of degradation of 0.06 mM MO by NaBH_4_ in the presence of 100 µL of each alloy NPs.

The reaction times were 80 s, 60 s, respectively. The size of particles decreased obviously, but catalytic performance was enhanced. It meant the physical treatment can promote the crystalline converting in growth process. The smaller sized Pd-Cu alloy NPs increase the contact with dye molecules, accelerated the degradation completely. The inset image in Figure 7b–e, it showed two kinds of Pd-Cu alloy NPs (Pd_5_Cu_1_, Pd_7_Cu_1_), the performance in catalytic MO molecule while using high intensity focused ultrasound treated Pd-Cu NPs. According to details, the relative degradation reaction just needed 30 s, 18 s, 20 s, 9 s, respectively. The apparent rate constants (K_app_) for all samples were presented in Appendix A. These results clearly showed that the activity of the alloy NPs was treated under high intensity focused ultrasound was much higher than that of the untreated alloy NPs. The Pd_7_Cu_1_ (10 MPa, 4000 W) alloy NPs showed the highest reaction rate (K_app_ = 356.7 × 10^−3^ s^−1^) among all the catalyst, which was also nearly 7 times than original NPs. Before handle by ultrasonic, the degradation reaction of Pd_5_Cu_1_ and Pd_7_Cu_1_ alloy NPs need 80 s and 60 s, respectively. It showed a strong size effect in physical field and ultrasound chemistry.

## 3. Experimental Section

**Materials.** Palladium (II) hexafluoroacetylacetonate (Pd(C_5_HF_6_O_2_)_2_), Copper acetylacetonate (C_1_OH_14_CuO_4_, 97%), NaBH_4_ (95%), glutathione (GSH, 99%), sodium hydroxide (NaOH, 95%), Absolute alcohol (100%). Methyl orange (MO), methylene blue (MB), rhodamine B (RhB), the 3,3′,5,5′-tetramethylbenzidine (TMB, >99%), H_2_O_2_ (30%, extra pure) was supplied by Aladdin Reagent CO., Ltd. (Shanghai, China). Spherical cavity focused transducer (Chongqing Haifu Technology Co., Ltd., Chongqing, China). All chemicals were used as received without further purification.

**Synthesis of Pd-Cu Alloy Nanoparticles (NPs).** A single step approach was employed for the synthesis of Pd_x_Cu_y_ alloy NPs with varying compositions, where x = 1, 2, 3, 5, 7 and y = 1, 2, 3, 5, 7. Typically, fill a beaker with equal proportions of 20 mL ultrapure water and 20 mL anhydrous ethanol. 0.02 mmol of Pd(C_5_HF_6_O_2_)_2_, 0.02 mmol of Cu(acac)_2_ and 0.06 mmol of GSH were respectively added to the above beaker. After ultrasonic dissolving for 5 min in an ultrasonic cleaner, the beaker was placed on a magnetic stirrer, and the speed was adjusted to about 400 rpm. Then the pH of the solution was adjusted with NaOH (2 M). Observe the solution color change, adjust to about 11.5 ± 0.5. Weigh and dissolve 0.02 mmol of NaBH_4_ in 1 mL ultrapure water, then add 1 mL of NaBH_4_ drop by drop into the above beaker, continue the reaction for 3 h, get a transparent light green solution. 1 mL sample was added to 2 mL centrifuge tube, and 1 mL isopropyl alcohol was added to the above centrifuge tube, and the bottom precipitate was obtained for other physical or optical characterization.

**Sample Characterization.** The crystal and purity of as-synthesized alloy NPs, was characterized by powder X-ray diffraction (XRD) patterns recorded on a Bruker D8 Advance X-ray diffractometer by using Cu K as a radiation source at room temperature. Energy dispersive X-ray (EDX) spectrum and EDX elemental mapping were obtained using an FEI Tecnai G2 F20 electron microscope with the operation voltage of 200 kV. The Ultraviolet-visible-near infrared (UV-*vis*-NIR) diffuse reflectance spectra were employed to investigate the optical properties of different samples on a UV-3600i Plus UV-*vis* spectrometer. Firstly, the Pd-Cu alloy NPs were centrifuged and cleaned twice. The centrifuged product was dissolved in deionized water. A drop of the target product was dropped on the copper net to form water droplets. Let the sample dry naturally, then transmission electron microscopy (TEM), high angle circular dark field scanning transmission electron microscope (HAADF-STEM) were performed. By changing the pressure and power of high intensity focused ultrasound applied to the alloy NPs, to obtain samples for further testing.

**Catalytic Study. Catalytic Degradation of Dyes.** In order to measure the dye degradation activity of the catalyst, we have chosen MO, RhB and MB dyes as the standard and Pd_7_Cu_1_ alloy NPs as the catalyst (the most active sample of all the reactions). In this experiment, 2.5 mL of aqueous solution of dye (4 × 10^−5^ M) and 0.1 mL of 0.2 M NaBH4 were placed in a beaker. Then, 100 µL of aqueous solution containing 0.1 mg/ mL of catalyst was added to the above solution in the beaker and the UV-vis spectra were recorded as a function of time. For understanding the effect of the amount of catalyst on the reaction rate, the amount of catalyst was varied from 0.2~0.8 mg/mL; where as other parameters for dye and NaBH4 were kept constant (2.5 mL of 1 × 10^−4^ mM MO and 0.1 mL of 0.2 M NaBH4). To know the effect of concentration of dye on the rate of reaction, the concentration of MO was varied from 0.4 to 1.0 × 10^−4^ mM; however, the amounts of catalyst and NaBH4 were kept con-stant (0.2 mg and 0.1 mL of 0.2 M, respectively). 

**Catalytic Reduction of Nitro Compounds**. In order to study the activity of Pd7Cu1 alloy NPs as efficient catalyst, the reductions of 4-NP were chosen as model reactions. Catalytic reduction reaction study of 4-NP was similar to that described above. In this process, we added 2.5 mL of a solution of 4-NP (0.04–0.1 mM) to the beaker, then, 0.1 mL of 0.2 M NaBH4 were placed in the beaker, the amounts of catalyst were kept constant (100 µL, 0.2 mg/mL). For studying the effect of the amount of catalyst, the amount of catalyst was varied (100 µL, 0.2 mg–0.8 mg/mL), however, volume and concentration of 4-NP and NaBH4 were kept constant (2.5 mL of 0.1 mM and 0.1 mL of 0.2 M, respectively).

**Cell Culture and Cell Viability Assay.** 239 T cell, HACAT cell, IMEF cell. All cells were maintained in a humidified incubator at a humidity of 37 and under a 5% CO_2_ atmosphere. Cell viability and apoptosis were determined using the Muse™ Cell Analyzer (PB4455ENEU, Millipore Co., Billerica, MA, USA). The required cells were stained according to the method explained in the experimental instructions, and then the corresponding measurements were per-formed using the fluorescence signal analysis equipment. The prepared cells were suspended in culture medium to a final cell concentration of 1 × 10^−5^ cells mL^−1^. For the cell viability analysis, 20 µL of suspended cells were stained with 380 µL of Muse™ Count & Viability Reagent (MCH100102, Millipore Co.) and incubated for 30 min at room temperature; each cell was subsequently analyzed. For apoptosis and necrosis analysis, 100 µL of suspended cells were stained with 100 µL of Muse™ Annexin V & Dead Cell Reagent (MCH100105, Millipore Co.) and incubated for 30 min at room temperature; each cell was then analyzed. After confluence of the cells was achieved, the cells were exposed to various concentrations (0, 2, 5, 12.5, 25, and 50 µg mL^−1^) of Pd_5_Cu_1_, Pd_7_Cu_1_, Pd_1_Cu_5_ and Pd_1_Cu_7_ alloy NPs for 2 days. In investigate the dose-dependent effect of cells on the alloy NPs, aqueous solutions of Pd_5_Cu_1_, Pd_7_Cu_1_, Pd_1_Cu_5_ and Pd_1_Cu_7_ alloy NPs at different concentrations (0, 25, 50 µg mL^−1^) were applied to each cell line, and then the survival of cells in different growth cycles (min) was observed.

**Standard Colorimetric of H_2_O_2_.** Colorimetric determination of H_2_O_2_ in aqueous solution was performed at room temperature under optimal conditions. PBS buffer with a pH of about 7.2–7.4 was used in the experiment. First, 400 µL of neutral PBS buffer was added into the centrifuge tube, and then a certain amount of 100 µL Pd_1_Cu_7_ alloy NPs (most active composition) was added. In colorimetric detection mode, 100 µL of TMB (10 mM) was added to the test system. After that, H_2_O_2_ solution with the concentration of 4 mM to 14 mM was prepared with the serially diluted stock solution. After the reaction system was left for 5 min, UV-vis spectra were analyzed, and pictures were taken with a digital camera to observe the color change. To measure the enzyme parameters, we choice the various concentrations of substrates (TMB) (4~18 mM) to test. The kinetic parameter v = V_max_[S]/ (K_m_ + [S]) was determined according to the Michaelis-Menten function, where v is the initial reaction velocity, is the maximum reaction rate, [S] is the substrate concentration, K_m_ is the Michaelis-Menten constant. This constant is equal to the substrate concentration with a conversion rate of half V_max_ and represents the affinity of the enzyme. V_max_ was calculated as the molar change in UV absorbance according to the equation A = Ɛwa (A is absorbance, Ɛ is absorbance coefficient, l is path length, c is molar concentration), where Ɛ = 3.9 × 10^4^ M^−1^ cm^−1^ and l = 1 mm.

**The enhancement of the high intensity focus ultrasound effect on NPs.** In this work, a spherical cavity focused transducer (Chongqing Haifu Technology Co., Ltd., Chongqing, China) with an inner diameter of D = 480 mm and two open ends (opening aperture of d = 219 mm and the height of H = 427 mm) was used, which can cause serious bubble cavitation. On the inner surface of the transducer, there are 48 pieces of high-power piezoelectric ceramics. The transducer excites at a spherically symmetric intrinsic frequency of about 600 KHz. Using a signal generator, power amplifier (maximum output power is 100 KW), impedance matching box respectively generate information, amplify the signal and match the electrical impedance of the transducer. The water treatment module is used for degassing the aqueous solution during the experiment. By controlling the parameters of high intensity focus ultrasound equipment, acoustic cavitation through the power and intensity. The hydrostatic pressure of the solution is 0.1 MPa, 3 MPa, 7 MPa, 10 MPa, respectively. The power is 100 W, 1000 W, 2000 W, 3000 W, 4000 W, respectively. To study the catalytic reduction capacity of alloy NPs treated, MO was chosen as a reduction model in the presence of excess NaBH_4_. By adding 250 µL (0.06 mM) MO and 100 µL (0.1 M) NaBH_4_ into a 350 µL colorimetric dish, it is to observe the color change of the solution. The reduction reaction of alloy NPs treated with different key parameters was carried out at room temperature. The catalytic reduction of Mo was monitored by UV-vis absorption spectroscopy. 

## 4. Conclusions

In summary, we have demonstrated a one-step method for the synthesis of ultra-small and ultra-hydrophilic Pd-Cu alloy NPs at ambient atmosphere. It implied the high stability in storing and the UV-*vis* spectroscopic analysis, which is realized even past 30 days cultivating with water aqueous. The as-synthesized alloy NPs were ~10 nm in diameter. It exhibits the relative ultra-hydrophilic property due to the smaller size and GSH surfactant. Owing to the difference of electronegativity of Pd, Cu element and smaller size of the alloy NPs, which displayed higher efficient catalytic reduction of the degradation of organic molecules and-aromatic compounds. The different cell lines (IMEF, HACAT, 239 T Cell lines) exposed to Pd-Cu alloy NPs for 2 days exhibited highly active at concentrations of up to 50 µg mL^−1^, which indicated of excellent biocompatibility. By using typical TMB as chromogenic substrate, we demonstrate that the Pd-Cu alloy NPs possess peroxidase-like activity. It means this obtained NPs could be utilized in biocatalyst or detection analysis. The special treatment as we used high intensity focused ultrasound field, the physical and chemical functions could make alloys smaller than before and increase the surface area eventually. The enhancement in catalytic performance give expression to one efficient way to synthetic-approach to catalyst. According to above detailed information, this work provided a facile method for super-hydrophilic Pd-Cu alloy NPs as highly active and biocompatible catalyst for dyes degradation. And the designed strategy could be one effective supplement for chemical mechanism in new materials’ constructions.

## Data Availability

The data presented in this study are available on request from the corresponding author.

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
