# Peer review of "Bioinspired Pd-Cu Alloy Nanoparticles as Accept Agent for Dye Degradation Performances"

_ijms, 2022, doi:10.3390/ijms232214072_

Round 1
Reviewer 1 Report
The paper, titled Bioinspired Pd-Cu Alloy Nanoparticles as Accept Agent for Dye Degradation Performances, provides a very good overview of current knowledge regarding the substitution of binary metal nanoparticles in dye removal. However, in my opinion, in the introduction, the methods of synthesis of such nanoparticles should be separated and constitute a separate chapter. The currently known pathways for the synthesis of binary nanoparticles should be described in great detail. Also a very interesting aspect in my opinion would be to confront the methods of synthesis of nanoparticles with their structural and chemical properties. In my opinion, it would also be worthwhile to briefly cover other applications of nanoparticles. Please cite the following work: https://doi.org/10.3390/app10217571. In addition, I would like to point out that the work is prepared very carefully, the drawings are clear and of very good quality.
Overall, I think the work is worthy of publication in International Journal of Molecular Sciences after the Major Revision.
Reviewer 2 Report
In current work (Manuscript ID: ijms-1979714) Chen and his coworkers reported the fabrication of Pd-Cu alloy nanoparticles via the co-reduction of Cu(acac)2 (acac = acetylacetonate) and Pd(C5HF6O2)2 in the presence of sodium borohydride (NaBH4) and glutathione (GSH). They investigated that the colloidal solution of the Pd-Cu alloy nanoparticles maintains stability for 30 days. The Pd-Cu nanoparticles revealed biocompatibility in form of cell lines (IMEF, HACAT, and 239T) exposed to the colloidal solution (50 μg mL-1) for 2 days. Further, Pd-Cu nanoparticles revealed good catalytic performance for methyl orange (MO), rhodamine B (RhB), and methylene blue (MB) dyes degradation. In addition, Pd-Cu nanoparticles exhibited peroxidase-like activity in the 3, 3’, 5, 5’-Tetra-methylbenzidine (TMB) color test and revealed obvious difference in comparison to the individual metal materials. This work is well organized and has scientific value. I would like to recommend its publication after amendments as noted below:
1. In the introduction part, authors should also discuss about the role of Au-Cu alloys in catalysis and their biocompatibility. Why Au-Cu alloys not employed in current work? Authors should refer to the following references.
https://doi.org/10.1039/B817729P ; https://doi.org/10.3390/en14051278
2. There exists a lot of improper sentences like “For measuring dye degradation of a catalytic of a cata-lytic chosen MO, RhB and MB dyes as the standard Pd7Cu1 alloy NPs as the catalytic (most active composition).” Authors should carefully revise the whole manuscript and do corrections accordingly.
3. Authors are suggested to check the catalyst performance for chlorophenols especially the 2,4-Dichlorophenol which has strong chlorine carbon bonds and difficult to degrade.
4. In the results and discussion part, “The peaks of Pd-Cu alloy NPs at 38.2º, 44.2º, and 64.4º, respectively.” this sentence is incomplete.
5. In Figure 1c, the high intensity peaks should also be indexed.
6. Why the MO, RhB and MB dyes are selected as model pollutants. Authors should mention in the text.
7. Any reason for adding NaBH4 to the aqueous solution of dyes? How about degradation reactions in its absence? Authors should mention the reaction of NaBH4 with dye molecules.
8. The conclusion should be more concise and different from abstract.
9. The exists a lot of grammatical mistakes, typo-errors and space problems. Authors should carefully revise the whole manuscript. Special attention should be given to the expression of sentences.
10. References should be consistent according to the format. Check reference # 45, 49, 77 and all others.
The abbreviation of ACS Sustainable Chemistry and Engineering is “ACS Sustain. Chem. Eng”. Likewise, authors should confirm abbreviations of all other journals in the references part.

Reviewer 3 Report
The primary purpose of the work presented in the paper ijms-1979714 is degradation of commonly used dyes (methyl orange (MO), rhodamine B (RhB), and methylene blue (MB) with using of the Pd-Cu alloy. However, the use of a simple application of a Pt-Cu alloy is beyond the scope of this work. The investigated catalyst is a new material that can be used in many applications. One of them can be compared with the toxicity via cell viability in the form of NPs integrated growth with different cell lines. The detailed description of the preparation of the Pt-Cu catalyst is precise, so the presented work is related both to the described catalytic dye degradation process and contributes to the general use of nanoparticles in chemical processes. Thus, the manuscript should be approved for publication.
Round 2
Reviewer 1 Report
I accept paper in present form
Reviewer 2 Report
Authors carefully addressed the reviewers comments. Manuscript should be accepted in it's current form.